# Circulating Growth Differentiation Factor 15 Is Associated with Diabetic Neuropathy

**DOI:** 10.3390/jcm11113033

**Published:** 2022-05-27

**Authors:** Shao-Wen Weng, Wen-Chieh Chen, Feng-Chih Shen, Pei-Wen Wang, Jung-Fu Chen, Chia-Wei Liou

**Affiliations:** 1Department of Internal Medicine, Kaohsiung Chang Gung Memorial Hospital and Chang Gung University College of Medicine, Kaohsiung 833, Taiwan; wengsw99@cgmh.org.tw (S.-W.W.); chingjing@cgmh.org.tw (W.-C.C.); atlanta.shen@gmail.com (F.-C.S.); wangpw@cgmh.org.tw (P.-W.W.); 0722cjf@cgmh.org.tw (J.-F.C.); 2Center for Mitochondrial Research and Medicine, Kaohsiung Chang Gung Memorial Hospital, Kaohsiung 833, Taiwan; 3School of Medicine, College of Medicine, Chang Gung University, Taoyuan 333, Taiwan; 4Department of Neurology, Kaohsiung Chang Gung Memorial Hospital and Chang Gung University College of Medicine, Kaohsiung 833, Taiwan

**Keywords:** growth differentiation factor 15, diabetic neuropathy, nerve conductive study, type 2 DM

## Abstract

Background: Growth differentiation factor (GDF15) is a superfamily of transforming growth factor-beta which has been suggested to be correlated with various pathological conditions. The current study aimed to investigate the predicted role of circulating GDF15 in diabetic metabolism characteristics and diabetic neuropathy. Methods: 241 diabetic patients and 42 non-diabetic subjects were included to participate in the study. The plasma GDF15 levels were measured using ELISA. Chronic kidney disease and albuminuria were defined according to the Kidney Disease: Improving Global Outcomes (KDIGO) guideline. The nerve conductive study (NCS) was performed with measurement of distal latency, amplitude, nerve conduction velocity (NCV), H-reflex, and F-wave studies. Results: The diabetic group had a significantly higher prevalence of chronic kidney disease and higher plasma GDF15 level. After adjusting for age and BMI, GDF15 was significantly positively correlated with waist circumference (*r* = 0.332, *p* = <0.001), hip circumference (*r* = 0.339, *p* < 0.001), HbA1c (*r* = 0.302, *p* < 0.001), serum creatine (*r* = 0.146, *p* = 0.017), urine albumin/creatinine ratio (*r* = 0.126, *p* = 0.040), and HOMA-IR (*r* = 0.166, *p* = 0.007). As to NCS, GDF15 was significantly correlated with all latency and amplitude of sensory and motor nerves, as well as F-wave and H-reflex latencies. The area under the curve (AUC) in predicting tibial motor nerve neuropathy (MNCV) in all subjects and in the diabetic group for GDF15 was 0.646 (*p* = 0.001) and 0.610 (*p* = 0.012), respectively; for HbA1c was 0.639 (*p* = 0.001) and 0.604 (*p* = 0.018), respectively. Predicting ulnar sensory nerve neuropathy for GDF15 was 0.639 (*p* = 0.001) and 0.658 (*p* = 0.001), respectively; for HbA1c was 0.545 (*p* = 0.307) and 0.545 (*p* = 0.335), respectively. Predicting median sensory nerve neuropathy for GDF15 was 0.633 (*p* = 0.007) and 0.611 (*p* = 0.032), respectively; for HbA1c was 0.631 (*p* = 0.008) and 0.607 (*p* = 0.038), respectively. Predicting CKD for GDF15 was 0.709 (95% CI, 0.648–0.771), *p* < 0.001) and 0.676 (95% CI, 0.605–0.746), *p* < 0.001), respectively; for HbA1c was 0.560 (95% CI, 0.493–0.627); *p* = 0.080) and 0.515 (95% CI, 0.441–0.588); *p* = 0.697), respectively. Conclusions: We suggest that there is a significant association between the increased serum GDF-15 level and metabolic parameters and diabetic neuropathy. Plasma GDF15 may be an independent predictor of diabetic neuropathy.

## 1. Introduction

Diabetes mellitus (DM) is a major global health concern facing the general population, particularly, the elderly people [1,2]. The complications resulting from diabetic microvascular and macrovascular pathology can be devastating, potentially resulting in physical disability or death [3,4,5]. Such diabetic complications are often caused by chronic hyperglycemia, which may induce oxidative damage and chronic inflammation [6]. Several inflammatory biomarkers have previously been identified to be associated with these complications, including CRP, IL-6, TNF-a, and ICAM-1 [7,8,9,10,11,12].

Growth differentiation factor (GDF15), also known as macrophage inhibiting cytokine 1, is a superfamily of transforming growth factor-beta with anti-inflammatory properties. It is widely expressed in various tissues throughout the body, including the cardiovascular system, metabolic system, and nervous system. It has also been found to be associated with the aging process, with an increased expression in the elderly [13,14]. Meanwhile, the circulating GDF15 levels have been reported to be correlated with heart failure, cancer, renal dysfunction, diabetes mellitus, mitochondrial disease, and obesity [15,16]. Overexpression of GDF15 has been noted to protect against high-fat diet-induced obesity and glucose intolerance [17], while GDF15-knockout mice have been reported to gain body weight and store fat, concomitant with increased food intake [18]. As to diabetic complications, previous studies have demonstrated that serum GDF15 concentrations are related to cardiovascular disease and diabetic nephropathy and retinopathy [19,20,21,22,23], indicating its potential role as a novel biomarker.

Although there is currently a lack of insight into the role of GDF15 in diabetic peripheral neuropathy (DPN), previous studies have reported on several risk factors associated with DPN, including age [24], BMI [25,26], HbA1c [27,28], urine albumin/creatinine ratio (ACR) [29,30], and blood pressure [31,32]. The major pathophysiological mechanism of DPN is possibly related to hyperglycemia-mediated cell injury. Hyperglycemia induces inadequately functioning vascular supply, while mitochondria and lipid metabolism may impair neurotropic support which induces downstream oxidative stress, as well as mitochondrial dysfunction and inflammation, ultimately causing cellular dysfunction and death [33,34].

Based on the risk factors and the possible pathophysiological mechanisms of DPN, we speculate that GDF15 may be an effective early predictor of DPN. Therefore, the current study aimed to investigate the predicted role of plasma GDF15, as compared with traditional risk factors, in DPN between diabetic and non-diabetic subjects.

## 2. Subjects and Methods

Two hundred eighty-three subjects were recruited to participate in the study, mainly from the Metabolism Clinic in Kaohsiung Chang Gung Memorial Hospital (KCGMH), Taiwan. Subjects included in this study were over 30 years old, including 241 T2DM patients and 42 non-diabetic subjects. Diabetes status was defined according to the diagnostic criteria of the American Diabetic Association with FPG ≥ 126 mg/dL or glycated hemoglobin A1c HbA1c ≥ 6.5%. All procedures adhered to the Declaration of Helsinki. The study was approved by the Ethics Committee of Chang Gung Memorial Hospital Institutional Review Board (IRB approval no. 201900948B0C601), and all individuals gave their informed consent.

## 3. Baseline Clinical and Laboratory Measurements

Demographic data, including body mass index, waist circumstance, systolic and diastolic blood pressure, and clinical laboratory parameters were obtained at baseline, including glycated hemoglobin A_1c_(HbA_1c_), serum creatine and lipid profiles, fasting glucose, and urine albumin and creatine. The levels of serum hs-CPR and HOMA-IR were detected by a biochemical auto analyzer (Beckman CX-7 Biochemical Autoanalyzer, Brea, CA, USA). Dyslipidemia was defined as a total cholesterol level of more than 200 mg/dL and/or triglyceride level of more than 150 mg/dL, or treatment with lipid-lowering agents.

Albuminuria was assessed by measuring the urinary albumin-to-creatinine ratio (UACR) in spot urine collected from the first voiding of urine in the morning. Chronic kidney disease and albuminuria were defined according to the Kidney Disease: Improving Global Outcomes (KDIGO) guideline [35]. Hypertension was defined as a systolic blood pressure of more than 140 mmHg and/or diastolic blood pressure of more than 90 mmHg or having a prescription for anti-hypertensive medicine.

## 4. Methods for Determination of GDF15 (ELISA)

The circulating levels of GDF15 were determined by the RayBio^®^ Human GDF15 ELISA Kit (RayBiotech Inc., Norcross, GA, USA) according to the manufacturer’s protocol. Briefly, 100 μL of each standard and sample was added to the appropriate wells. After 2.5 h of incubation, the solution was discarded and washed four times with washing solution. Then, 100 μL of biotinylated antibody was added to each well and incubated for 1 h at room temperature. After four washes, 100 μL of streptavidin solution was added to each well and incubated for 45 min. After four washes, 100 μL of TMB one-step substrate reagent was added to each well and incubated at room temperature for 30 min in the dark. Finally, 50 μL of stop solution was added to each well and read the absorbance at 450 nm using a microplate reader immediately.

## 5. Nerve Conduction Studies

The nerve conductive study (NCS) was performed using Nicolet Viking machines and data were compared with reference values as previously reported [36]. Surface recording and stimulation were recorded using standard laboratory methods. The belly-tendon montage and supra-maximal stimulation were applied. The motor nerve NCS included the median, ulnar, tibial, and peroneal nerves, and the sensory nerve NCS included the median and ulnar nerves. The following attributes were measured: distal latency, amplitude (Amp), and nerve conduction velocity (NCV). The late responses study included an H-reflex study and median, ulnar, peroneal, and tibial F-wave studies.

## 6. Statistical Analysis

Quantitative variables were shown as the mean ± standard deviation (SD) or median with interquartile range (IQR) in descriptive analyses, while categorical data were represented as numbers and percentages. Continuous variables that were not normally distributed were logarithmically transformed to improve normality prior to analysis. For comparisons between groups, continuous variables were compared using Student’s *t*-tests, or the Mann–Whitney U test. Categorical variables were compared using chi-squared tests. Spearman correlation analysis was used to calculate the correlation coefficients between GDF15 and NCS parameters with partial correlation adjusted for age, B.W, B.H, and HbA1c and partial correlation adjusted for age and BMI between GDF15 and metabolic parameters. Linear regression analyses were performed to evaluate the influence of GDF15 on the NCV parameters, adjusting for age, B.W, B.H, and HbA1c. To evaluate the discriminating ability of our risk prediction model for chronic kidney disease and peripheral neuropathy patients, we calculated the area under the receiver operating characteristic (ROC) curve (AUC). We compared the AUCs of GDF15 and traditional risk factors. The optimal cut-off value was determined by Youden’s index. All statistical analyses were performed with the SPSS software (version 25.0). A two-sided *p* < 0.05 was considered to be statistically significant.

## 7. Results

### 7.1. Comparisons between DM and Non-DM Group

A total of 241 T2DM patients and 42 non-diabetic subjects were enrolled in this study. As shown in Table 1, the diabetic group had a significantly higher prevalence of chronic kidney disease; higher serum HbA1c, GDF15, and urine albumin/creatine ratio. The comparisons between the DM and non-DM groups are listed in Table 2. Patients with DM had longer latency in the sensory median nerve (*p* < 0.001), motor median (*p* < 0.001), ulnar (*p* = 0.039), and peroneal (*p* = 0.011) nerves; longer F-wave latency in median (*p* = 0.002), ulnar (*p* < 0.001), peroneal (*p* < 0.001), and tibial (*p* = 0.001) nerves; and longer H-reflex latency in tibial nerves (*p* = 0.003); had lower sensory amplitude in median and ulnar nerves (*p* = 0.039 and 0.021, respectively). As to the nerve conduction velocity (NCV), those with DM had lower sensory NCV values in median (*p* = 0.008) and lower motor NCV in median (*p* = 0.002), ulnar (*p* = 0.038), peroneal (*p* = 0.026), and tibial (*p* = 0.002) nerves.

### 7.2. The Risk Prediction Model for Nephropathy

We calculated the area under the receiver operating characteristic (ROC) curve to evaluate the effectiveness of our GDF15 prediction model for patients with chronic kidney disease (CKD). The risk prediction model was determined from known risk factors for CKD, including age, HbA1c, systolic blood pressure (SBP), and diastolic blood pressure (DBP). (Figure 1) The area under the curve (AUC) in all subjects and the diabetic group for GDF15 was 0.709 (95% CI, 0.648–0.771), *p* < 0.001 and 0.676 (95% CI, 0.605–0.746), *p* < 0.001, respectively; for age was 0.597 (95% CI, 0.531–0.663); *p* = 0.005 and 0.609 (95% CI, 0.537–0.680), *p* = 0.004, respectively; for SBP was 0.582 (95% CI, 0.515–0.648); *p* = 0.018 and 0.580 (95% CI, 0.508–0.652), *p* = 0.034, respectively; for DBP was 0.577 (95% CI, 0.510–0.644), *p* = 0.025 and 0.589 (95% CI, 0.516–0.661); *p* = 0.019, respectively; for HbA1c was 0.560 (95% CI, 0.493–0.627); *p* = 0.080 and 0.515 (95% CI, 0.441–0.588); *p* = 0.697, respectively. Furthermore, we selected a GDF15 cut-off value of 430 (pg/mL) to reach a sensitivity of 78% and specificity of 60% in all subjects and a cut-off value of 431 (pg/mL) to reach a sensitivity of 82% and specificity of 52% in the diabetic group, respectively.

### 7.3. Comparisons between Different Levels of GDF15

Comparisons between different levels of GDF15 are shown in Table 3. Subjects with a higher GDF15 level (≥530 pg/mL) had a significantly higher waist circumference (WC) (*p* < 0.001) and hip circumference (HC) (*p* = 0.007), while there was no significant difference of BMI between these two groups. Those with a higher GDF15 level had a significantly higher prevalence of DM (*p* < 0.001), CKD (*p* < 0.001) and hypertension (*p* = 0.014); higher serum HbA1c (*p* = 0.001), HOMA-IR (*p* = 0.003), creatinine (*p* = 0.001), and urine ACR (*p* < 0.001). As to the NCS results (Table 4), patients with a higher GDF15 had longer latency in the sensory median (*p* = 0.007) and ulnar (*p* < 0.001) nerves, motor median (*p* = 0.038), ulnar (*p* = 0.001), and peroneal (*p* = 0.048) nerves; longer F-wave latency in median (*p* = 0.004), ulnar (*p* < 0.001), peroneal (*p* = 0.004), and tibial (*p* = 0.004) nerves; and longer H-reflex latency in tibial nerves (*p* = 0.006); had lower sensory amplitude in ulnar nerves (*p* = 0.034). As to nerve conduction velocity (NCV), those with a higher GDF15 had lower sensory NCV values in median (*p* < 0.001) and ulnar (*p* < 0.001) nerves, and lower motor NCV in median (*p* = 0.019), ulnar (*p* = 0.018), peroneal (*p* = 0.013), and tibial (*p* = 0.006) nerves. After adjusting for age, B.W, B.H, and HbA1c, most of these parameters of NCS maintained significant differences between those with different levels of GDF15. Subjects with a higher GDF15 level still had longer latency in the sensory ulnar (*p* < 0.001) nerves and motor ulnar (*p* = 0.008) nerves; longer F-wave latency in ulnar (*p* < 0.001), peroneal (*p* = 0.008), and tibial (*p* = 0.007) nerves; and longer H-reflex latency in tibial nerves (*p* = 0.012); had lower sensory NCV values in median (*p* < 0.001) and ulnar (*p* < 0.001) nerves, lower motor NCV in peroneal (*p* = 0.034), and tibial (*p* = 0.029) nerves.

### 7.4. Correlation between GDF15 and Nerve Conductive Study

The correlation analysis used to test the influence of GDF15 on NCS and clinical factors is listed in Table 5 and Table 6, respectively. The significant statistical results (correlation coefficient, *p*-value) of NCS were as follows: sensory median nerve latency and NCV (*r* = 0.229, *p* = <0.001 and r = −0.224, *p* < 0.001), sensory ulnar nerve latency and NCV (*r* = 0.152, *p* = 0.012 and r = −0.188, *p* = 0.002), motor ulnar nerve latency and NCV (*r* = 0.178, *p* = 0.003 and r = −0.143, *p* = 0.055), motor tibial nerve latency and NCV(*r* = 0.168, *p* = 0.005 and r = −0.203, *p* = 0.001), motor peroneal NCV (*r* = −0.138, *p* = 0.023), F-wave latency in median nerve (*r* = 0.137, *p* = 0.022), F-wave latency in ulnar nerve (*r* = 0.192, *p* = 0.001), F-wave latency in peroneal nerve (*r* = 0.192, *p* = 0.005), F-wave latency in tibial nerve (*r* = 0.209, *p* = 0.001), H-reflex latency in tibial nerve (*r* = 0.205, *p* = 0.001), respectively. After controlling for age B.W and B.H, most NCS parameters remained statistically significant, except for sensory median latency, ulnar latency, and motor ulnar amplitude (*p* = 0.054, 0.052, and 0.075, respectively). With regards to the clinical factors, GDF15 demonstrated a significant positive correlation with WC (*r* = 0.254, *p* = <0.001), HC (*r* = 0.165, *p* = 0.005), HbA1c (*r* = 0.306, *p* < 0.001), serum creatine (*r* = 0.240, *p* < 0.001), urine ACR (*r* = 0.318, *p* < 0.001); while GDF15 was negatively correlated with eGFR (*r* = −0.275, *p* < 0.001). After controlling for age and BMI, all these parameters remained statistically significant, while HOMA-IR also reached statistical significance (*r* = 0.166, *p* = 0.007).

### 7.5. The Risk Prediction Model for Neuropathy

To verify the serum GDF15 prediction model with conventional neuropathy risk factors, we evaluated the different AUC estimates among these factors. The well-known risk factors for neuropathy include age, HbA1c, urine ACR, SBP, and DBP. The area under the curve (AUC) in the prediction of tibial motor nerve neuropathy (MNCV) in all subjects (Figure 2a) and the diabetic group (Figure 2b) for GDF15 was 0.646 (*p* = 0.001) and 0.610 (*p* = 0.012), respectively; for age was 0.540 (*p* = 0.342) and 0.557 (*p* = 0.193), respectively; for HbA1c was 0.639 (*p* = 0.001) and 0.604 (*p* = 0.018), respectively; for urine ACR was 0.579 (*p* = 0.060) and 0.557 (*p* = 0.193), respectively; for SBP was 0.498 (*p* = 0.969) and 0.496 (*p* = 0.923), respectively; for DBP was 0.481 (*p* = 0.644) and 0.476 (*p* = 0.589), respectively. Furthermore, we selected a GDF15 cut-off value of 429 (pg/mL) to reach a sensitivity of 82% and specificity of 47% in all subjects and a cut-off value of 717 (pg/mL) to reach a sensitivity of 57% and specificity of 67% in the diabetic group, respectively. As to sensory nerve neuropathy, GDF15 is a significant predictor of ulnar and median nerve neuropathy (SNCV). The area under the curve (AUC) in predicting ulnar sensory nerve neuropathy in all subjects (Figure 3a) and in the diabetic group (Figure 3b) for GDF15 was 0.639 (*p* = 0.001) and 0.658 (*p* = 0.001), respectively; for age was 0.557 (*p* = 0.192) and 0.576 (*p* = 0.105), respectively; for HbA1c was 0.545 (*p* = 0.307) and 0.545 (*p* = 0.335), respectively; for urine ACR was 0.557 (*p* = 0.193) and 0.582 (*p* = 0.081), respectively; for SBP was 0.452 (*p* = 0.272) and 0.419 (*p* = 0.086), respectively; for DBP was 0.426 (*p* = 0.091) and 0.397 (*p* = 0.028), respectively. Furthermore, we selected a GDF15 cut-off value of 489 (pg/mL) to reach a sensitivity of 78% and specificity of 54% in all subjects and a cut-off value of 556 (pg/mL) to reach a sensitivity of 79% and specificity of 56% in the diabetic group, respectively. The area under the curve (AUC) in predicting median sensory nerve neuropathy in all subjects (Figure 4a) and in the diabetic group (Figure 4b) for GDF15 was 0.633 (*p* = 0.007) and 0.611 (*p* = 0.032), respectively; for age was 0.587 (*p* = 0.078) and 0.609 (*p* = 0.035), respectively; for HbA1c was 0.631 (*p* = 0.008) and 0.607 (*p* = 0.038), respectively; for urine ACR was 0.546 (*p* = 0.354) and 0.532 (*p* = 0.538), respectively; for SBP was 0.577 (*p* = 0.117) and 0.559 (*p* = 0.250), respectively; for DBP was 0.497 (*p* = 0.957) and 0.479 (*p* = 0.687), respectively. Furthermore, we selected a GDF15 cut-off value of 552 (pg/mL) to reach a sensitivity of 76% and specificity of 57% in all subjects and a cut-off value of 552 (pg/mL) to reach a sensitivity of 79% and specificity of 50% in the diabetic group, respectively.

## 8. Discussion

This novel study demonstrates the association between the serum GDF15 level and peripheral neuropathy, as determined by a nerve conductive study. Here, we examined the amplitude and conduction velocity of the nerves in the lower extremities and correlated these with GDF15 levels and different metabolic parameters. Our study results revealed that patients with T2DM had a higher serum GDF15 level and prolonged motor and sensory nerve latency, as well as F-wave and H-reflex latency; in addition, patients had reduced sensory amplitude and slower SNCV and MNCV. Furthermore, subjects with a higher GDF15 level had significantly higher WC, HC, higher serum HbA1c, creatinine, and urine ACR, as well as a significantly higher prevalence of T2DM, CKD, and hypertension. These associations remained significant after adjusting for the confounding risk factors. Taken together, our results indicate that serum GDF15 is a strong predictor of tibial motor nerve neuropathy (MNCV) and ulnar and median sensory nerve neuropathy (SNCV) in diabetic patients; indeed, GDF15 is a superior predictive marker for these neuropathies as compared to the risk factors of age, HbA1c, urine ACR, SBP, and DBP.

GDF15, also known as macrophage inhibiting cytokine 1, is a superfamily of transforming growth factor-beta which is widely expressed in almost all tissues, including cardiovascular, metabolic, and nerve systems, while expression increases with age [14]. In addition, diabetic and obese subjects exhibit elevated serum levels of GDF15, which is correlated with many metabolic parameters [37]. Consistent with previous studies, we herein found that the serum GDF15 level is associated with higher WC, HC, HbA1c, and a higher prevalence of T2DM and hypertension. Regarding diabetic complications, several reports have demonstrated the role of serum GDF15 in diabetic nephropathy and have suggested that GDF15 is a predictor of diabetic kidney disease [38,39,40,41,42]. Furthermore, recent studies have described the association between serum GDF15 and diabetic retinopathy both in T2DM [22,23] and T1DM [39], reporting that the plasma GDF15 level is a significant predictor of diabetic retinopathy, while there is a trend between increased plasma GDF15 level and severity of retinopathy in T2DM. While this study did not investigate diabetic retinopathy in particular, we indeed demonstrate that circulating GDF15 is elevated in subjects with diabetic nephropathy and is positively correlated with urine ACR and negatively correlated with eGFR.

With regards to diabetic peripheral neuropathy (DPN), a previous study identified a higher serum GDF15 level in T1DM with peripheral neuropathy, which was defined by symptoms reported by a patient with abnormal touch sensation. In our study, we evaluated nerve conductivity as correlated with the circulating level of GDF15. We found that serum GDF15 is a significant predictor of DPN both in motor and sensory NCV. Further, we identified a positive correlation between the circulating GDF15 level and nerve latency, and a negative correlation with nerve amplitude and NCV after correcting for age, B.W, B.H, and HbA1c. Collectively, our study results reveal a significant association between serum GDF15 and peripheral neuropathy, as defined by NCV in subjects with T2DM.

Previous studies have suggested several risk factors associated with DPN, including age [24], BMI [25,26], HbA1c [27,28], urine ACR [29,30], and blood pressure [31,32]. Here, we established a risk prediction model based on circulating GDF15 which offers a superior method of identifying T2DM patients with and without DPN as compared to the abovementioned risk factors, demonstrating an effective diagnostic accuracy. However, the possible mechanism of the GDF15 on DPN in subjects with T2DM awaits further investigation.

GDF15 may provide the dual effects of pro-inflammatory and anti-inflammatory function in the development and progression of atherosclerosis, depending on the different progression stage and pathophysiological environment [43,44]. GDF15 is produced in activated macrophages under a pro-inflammatory status, vascular injury, and oxidative stress in human endothelial cells and in vascular smooth muscle cells [43]. Meanwhile, GDF15 can inhibit polymorphonuclear leukocyte recruitment by direct interference with chemokine signaling and integrin activation [45], which may play a compensatory role to counter-regulate the progression of vascular inflammation. The possible mechanisms by which GDF15 may be implicated in the pathogenesis of diabetic microvascular disease could involve oxidative stress, endothelial function [37,45], and immune processes [46,47], all of which are key pathogenic pathways of diabetic neuropathy. Furthermore, previous studies demonstrated that GDF15 may play a protective role in dopamine neurons. The possible mechanisms involved in reduced oxidative stress and inflammatory responses are related to PI3K/Akt and Akt/mTOR pathways [48,49]. In GDF15 knockout mice, a lack of GDF15 aggravates neuron loss and exogenous GDF15 promotes the survival of dopaminergic neurons [50,51]. Taken together, these make GDF15 to be a possible early biomarker for predicting diabetic neuropathy.

## 9. Study Limitations

The strengths of this study included a comprehensive nerve conductive study and clinical assessment; however, we should recognize some limitations in our study. The sample size for logistic regression models used for AUC was limited due to the small number of diabetics/and especially diabetic nephropathy subjects. Despite the adjustments made, residual confounding might remain due to the observational nature of the study. Spurious results might occur just by chance due to the multiple testing–alpha inflation. Our study population only consisted of subjects of Asian ethnicities, while the patient number was relatively limited. Thus, further validation of the risk prediction model in other populations of different ethnicities may be required. GDF15 level is associated with age, and age-specific comparisons in a future large cohort study are also recommended. A long-term longitudinal study may be necessary to further identify GDF15 as it is involved in diabetic neuropathy progression.

## 10. Conclusions

In the current study, we identified a significant association between increased serum levels of GDF15 and metabolic parameters and diabetic neuropathy; thus, plasma GDF15 may be an effective independent predictor of diabetic neuropathy. We suggest that further investigation is necessary to clarify the underlying mechanisms at play with GDF15 and DN.

## Figures and Tables

**Figure 1 jcm-11-03033-f001:**
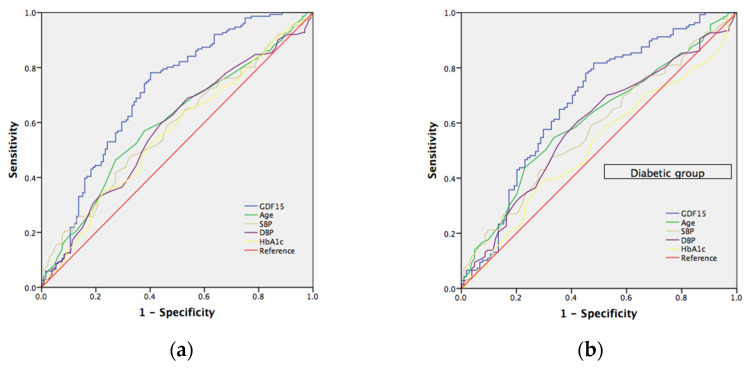
(**a**) Receiver operating characteristic (ROC) curves and area under the curve (AUC) in predicting chronic kidney disease. Blue line: GDF15 risk score (AUC = 0.709; *p* < 0.001); green line: age risk model (AUC = 0.597; *p* = 0.005); yellowish-green line: systolic blood pressure risk model (AUC = 0.582; *p* = 0.018); purple line: diastolic blood pressure risk model (AUC = 0.577; *p* = 0.025). yellowish line: HbA1c risk model (AUC = 0.560; *p* = 0.080). The red line represents the reference line. (**b**) Receiver operating characteristic (ROC) curves and area under the curve (AUC) in predicting chronic kidney disease in diabetic group. Blue line: GDF15 risk score (AUC = 0.676; *p* < 0.001); green line: age risk model (AUC = 0.609; *p* = 0.004); yellowish-green line: systolic blood pressure risk model (AUC = 0.580; *p* = 0.034); purple line: diastolic blood pressure risk model (AUC = 0.589; *p* = 0.019). yellowish line: HbA1c risk model (AUC = 0.515; *p* = 0.697). The red line represents the reference line.

**Figure 2 jcm-11-03033-f002:**
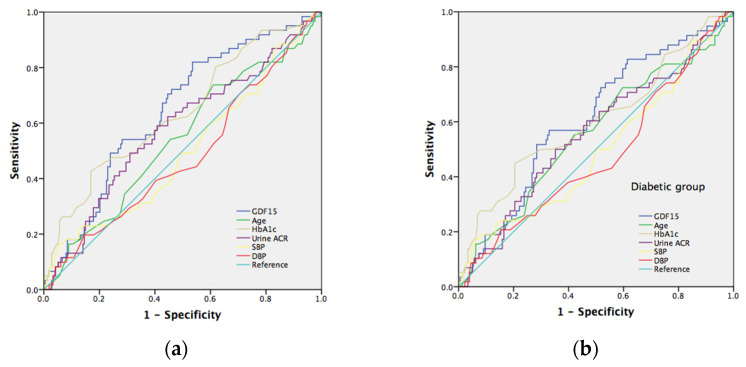
(**a**) Receiver operating characteristic (ROC) curves and area under the curve (AUC) in predicting tibial motor neuropathy. Blue line: GDF15 risk score (AUC = 0.646; *p* = 0.001); green line: age risk model (AUC = 0.540; *p* = 0.342); yellowish-green line: HbA1c risk model (AUC = 0.639; *p* = 0.001); purple line: urine albumin/creatine risk model (AUC = 0.579; *p* = 0.060). Yellowish line: systolic blood pressure risk model (AUC = 0.498; *p* = 0.969). Red line: urine diastolic blood pressure risk model (AUC = 0.481; *p* = 0.644); light blue line represents the reference line. (**b**) Receiver operating characteristic (ROC) curves and area under the curve (AUC) in prediction of tibial motor neuropathy. Blue line: GDF15 risk score (AUC = 0.610; *p* = 0.012); green line: age risk model (AUC = 0.557; *p* = 0.193); yellowish-green line: HbA1c risk model (AUC = 0.604; *p* = 0.018); purple line: urine albumin/creatine risk model (AUC = 0.557; *p* = 0.193). Yellowish line: systolic blood pressure risk model (AUC = 0.496; *p* = 0.923). Red line: urine diastolic blood pressure risk model (AUC = 0.476; *p* = 0.589); light blue line represents the reference line.

**Figure 3 jcm-11-03033-f003:**
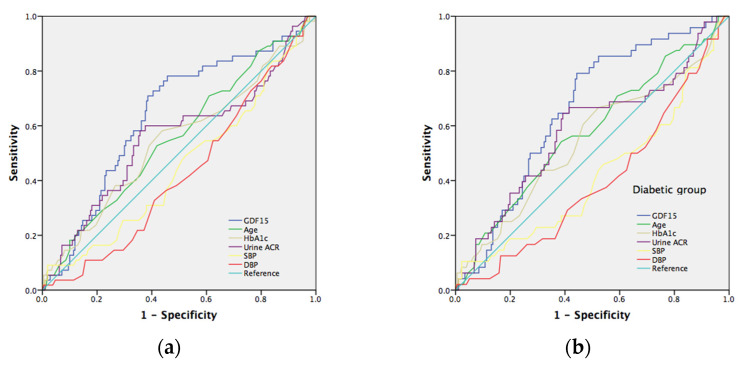
(**a**) Receiver operating characteristic (ROC) curves and area under the curve (AUC) in predicting ulnar nerve sensory neuropathy. Blue line: GDF15 risk score (AUC = 0.639; *p* = 0.001); green line: age risk model (AUC = 0.557; *p* = 0.192); yellowish-green line: HbA1c risk model (AUC = 0.545; *p* = 0.307); purple line: urine albumin/creatine risk model (AUC = 0.557; *p* = 0.193). Yellowish line: systolic blood pressure risk model (AUC = 0.452; *p* = 0.272). Red line: urine diastolic blood pressure risk model (AUC = 0.426; *p* = 0.091); light blue line represents the reference line. (**b**) Receiver operating characteristic (ROC) curves and area under the curve (AUC) in prediction of ulnar nerve sensory neuropathy. Blue line: GDF15 risk score (AUC = 0.658; *p* = 0.001); green line: age risk model (AUC = 0.576; *p* = 0.105); yellowish-green line: HbA1c risk model (AUC = 0.545; *p* = 0.335); purple line: urine albumin/creatine risk model (AUC = 0.582; *p* = 0.081). Yellowish line: systolic blood pressure risk model (AUC = 0.419; *p* = 0.086). Red line: urine diastolic blood pressure risk model (AUC = 0.397; *p* = 0.028); light blue line represents the reference line.

**Figure 4 jcm-11-03033-f004:**
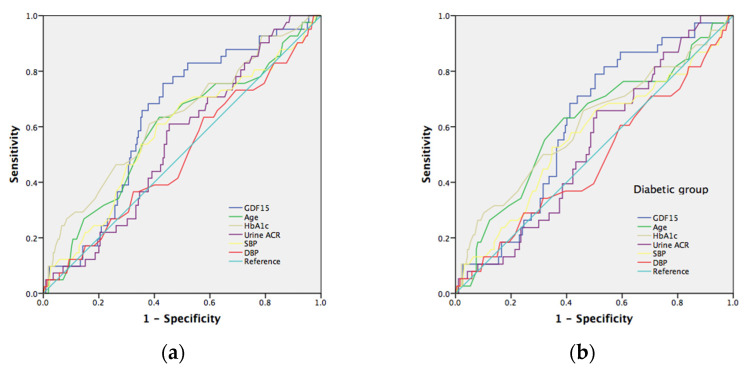
(**a**)**.** Receiver operating characteristic (ROC) curves and area under the curve (AUC) in prediction of median nerve sensory neuropathy. Blue line: GDF15 risk score (AUC = 0.633; *p* = 0.007); green line: age risk model (AUC = 0.587; *p* = 0.078); yellowish-green line: HbA1c risk model (AUC = 0.631; *p* = 0.008); purple line: urine albumin/creatine risk model (AUC = 0.546; *p* = 0.354). Yellowish line: systolic blood pressure risk model (AUC = 0.577; *p* = 0.117). The red line: urine diastolic blood pressure risk model (AUC = 0.497; *p* = 0.957); light blue line represents the reference line. (**b**) Receiver operating characteristic (ROC) curves and area under the curve (AUC) in predicting median nerve sensory neuropathy. Blue line: GDF15 risk score (AUC = 0.611; *p* = 0.032); green line: age risk model (AUC = 0.609; *p* = 0.035); yellowish-green line: HbA1c risk model (AUC = 0.607; *p* = 0.038); purple line: urine albumin/creatine risk model (AUC = 0.532; *p* = 0.538). Yellowish line: systolic blood pressure risk model (AUC = 0.559; *p* = 0.250). Red line: urine diastolic blood pressure risk model (AUC = 0.479; *p* = 0.687); light blue line represents the reference line.

**Table 1 jcm-11-03033-t001:** Comparisons between DM and non-DM.

	DM	Non-DM	*p*-Value
Number	241	42	
Age (years)	64.7 ± 8.9	68.9 ± 9.3	0.006
Men, n (%)	144 (59.8)	21 (50.0)	0.237
B.H (cm)	162 ± 8.5	159 ± 7.2	0.030
B.W (kg)	69.3 ± 12.8	63.5 ± 9.6	0.001
BMI (kg/m^2^)	26.3 ± 3.7	25.2 ± 3.1	0.033
WC (cm)	85.4 ± 10.2	78.8 ± 8.9	<0.001
HbA1c (%)	7.23 ± 1.0	5.78 ± 0.28	<0.001
Creatinine	1.03 ± 0.50	0.93 ± 0.32	0.196
eGFR (mL∙min^−1^1.73 m^−2^)	74.3 ± 23.8	75.2 ± 17.3	0.765
Urine ACR (mg/g)	30.5 (10.3–139.9)	11.4 (6.25–17.2)	<0.001
GDF15 (pg/mL)	606 (344–1127)	185 (144–326)	<0.001
GDF15 > 530 pg/mL, n (%)	121 (50.2)	4 (9.5)	<0.001
Hyperlipidemia, n (%)	182 (75.5)	26 (61.9)	0.065
HTN, n (%)	180 (74.7)	33 (78.6)	0.590
CKD, n (%)	137 (56.8)	14 (30.3)	0.005

Quantitative data are presented as the mean ± SD or median (IQR). B.H: body heigh; B.W: body weight; BMI: body mass index; WC: waist circumference; HbA1c: glycated hemoglobin; eGFR: estimated glomerular filtration rate; ACR: albumin-to-creatinine ratio; HTN: hypertension; CKD: chronic kidney disease.

**Table 2 jcm-11-03033-t002:** Comparisons between DM and non-DM in nerve conduction study.

	DM	Non-DM	*p*-Value
Sensory			
**Median nerve**	N = 229	N = 42	
Latencies (ms)	1.33 ± 0.38	1.19 ± 0.14	<0.001
Amp, µV	36.4 ± 20.1	43.2 ± 17.9	0.039
NCV, m/s	54.6 ± 9.2	58.6 ± 7.1	0.008
**Ulnar nerve**	N = 230	N = 42	
Latencies (ms)	2.34 ± 0.35	2.28 ± 0.29	0.239
Amp, µV	25.8 ± 13.0	31.2 ± 14.6	0.021
NCV, m/s	51.8 ± 6.4	52.8 ± 6.3	0.395
Motor			
**Median nerve**	N = 240	N = 42	
Latencies (ms)	4.45 ± 1.25	3.70 ± 0.40	<0.001
Amp, µV	9.14 ± 2.5	9.41 ± 2.3	0.5
NCV, m/s	52.3 ± 4.3	54.5 ± 4.2	0.002
**Ulnar nerve**	N = 241	N = 42	
Latencies (ms)	2.92 ± 0.41	2.78 ± 0.33	0.039
Amp, µV	9.12 ± 2.11	9.39 ± 2.00	0.45
NCV, m/s	53.4 ± 4.6	55.3 ± 4.3	0.038
**Peroneal nerve**	N = 231	N = 41	
Latencies (ms)	3.92 ± 0.61	3.72 ± 0.41	0.011
Amp, mV	4.55 ± 2.1	4.88 ± 2.1	0.359
NCV, m/s	44.8 ± 5.1	46.7 ± 5.2	0.026
**Tibial nerve**	N = 235	N = 41	
Latencies (ms)	4.00 ± 0.67	3.94 ± 0.74	0.6
Amp, mV	10.87 ± 4.4	12.22 ± 4.2	0.067
NCV, m/s	44.2 ± 4.5	46.6 ± 4.5	0.002
F-wave			
**Median nerve**	N = 236	N = 41	
Latencies (ms)	28.3 ± 3.0	26.7 ± 3.6	0.002
**Ulnar nerve**	N = 238	N = 41	
Latencies (ms)	27.9 ± 3.0	26.2 ± 2.0	<0.001
**Peroneal nerve**	N = 186	N = 31	
Latencies (ms)	50.0 ± 4.9	46.9 ± 3.4	<0.001
**Tibial nerve**	N = 232	N = 40	
Latencies (ms)	50.3 ± 4.9	47.7 ± 3.6	0.001
H-reflex			
**Tibial nerve**	N = 227	N = 40	
Latencies (ms)	32.3 ± 3.0	30.8 ± 2.4	0.003

Amp: amplitude; NCV: nerve conduction velocity.

**Table 3 jcm-11-03033-t003:** Comparisons between high and low GDF level.

GDF15 Level	<530	≥530	*p*-Value
Number	142	141	
Age (years)	64.6 ± 9.5	66.1 ± 8.6	0.185
Men, n (%)	77 (54.2)	88 (62.4)	0.163
B.H (cm)	161.5 ± 8.5	161.5 ± 8.3	0.997
B.W (kg)	68.3 ± 12.3	68.6 ± 12.8	0.804
BMI (kg/m^2^)	26.1 ± 3.5	26.2 ± 3.9	0.747
WC (cm)	82.1 ± 9.7	86.8 ± 10.3	<0.001
HC (cm)	89.1 ± 6.4	91.4 ± 7.8	0.007
HbA1c (%)	6.81 ± 1.0	7.22 ± 1.0	0.001
Creatinine	0.93 ± 0.33	1.11 ± 0.57	0.001
eGFR (mL∙min^−1^1.73m^−2^)	78.2 ± 21.5	70.0 ± 23.3	<0.001
Urine ACR (mg/g)	14.3 (6.7–69.5)	42.0 (12.5–177.5)	<0.001
0.68 (0.26–1.52) 0.94 (0.42–2.09) 1.99 (1.40–2.76) 2.22 (1.35–3.39) hsCRP (mg/L)	14.3 (6.7–69.5)	42.0 (12.5–177.5)	0.709
HOMA-IR	0.68 (0.26–1.52)	0.94 (0.42–2.09)	0.003
DM, n (%)	105 (73.9)	136 (96.5)	<0.001
Hyperlipidemia, n (%)	104 (73.2)	104 (73.8)	0.921
HTN, n (%)	98 (69.0)	115 (81.6)	0.014
CKD, n (%)	54 (38)	97 (68.8)	<0.001

Quantitative data are presented as the mean ± SD or median (IQR). B.H: body height; B.W: body weight; BMI: body mass index; WC: waist circumference; HC: hip circumference; HbA1c: glycated hemoglobin; eGFR: estimated glomerular filtration rate; ACR: albumin-to-creatinine ratio; hsCRP: high-sensitivity C-reactive protein; HOMA-IR: homeostatic model assessment for insulin resistance; HTN: hypertension; CKD: chronic kidney disease.

**Table 4 jcm-11-03033-t004:** Comparisons between GDF grading and nerve conduction velocity.

	GDF15 < 530	GDF15 ≥ 530	*p*	*p* *	*p* **
Sensory					
**Median nerve**	N = 136	N = 135			
Latencies (ms)	1.25 ± 0.38	1.36 ± 0.33	0.007	0.013	0.065
Amp, µV	39.1 ± 19.0	35.7 ± 20.7	0.158		
NCV, m/s	57.6 ± 7.2	52.8 ± 9.9	<0.001	<0.001	<0.001
**Ulnar nerve**	N = 140	N = 132			
Latencies (ms)	2.25 ± 0.27	2.42 ± 0.38	<0.001	<0.001	<0.001
Amp, µV	28.4 ± 14.0	24.8 ± 13.8	0.034	0.055	0.198
NCV, m/s	53.6 ± 5.5	50.3 ± 6.9	<0.001	<0.001	<0.001
Motor					
**Median nerve**	N = 142	N = 140			
Latencies (ms)	4.19 ± 1.11	4.49 ± 1.27	0.038	0.06	0.249
Amp, µV	9.41 ± 2.15	8.95 ± 2.74	0.113		
NCV, m/s	53.3 ± 4.3	52.1 ± 4.3	0.019	0.36	0.213
**Ulnar nerve**	N = 142	N = 141			
Latencies (ms)	2.82 ± 0.36	2.98 ± 0.43	0.001	0.001	0.008
Amp, mV	9.30 ± 1.97	9.02 ± 2.21	0.522		
NCV, m/s	54.5 ± 4.4	52.9 ± 4.7	0.018	0.019	0.113
**Peroneal nerve**	N = 141	N = 131			
Latencies (ms)	3.82 ± 0.55	3.96 ± 0.63	0.048	0.04	0.064
Amp, mV	4.71 ± 1.93	4.78 ± 2.28	0.37		
NCV, m/s	45.8 ± 4.6	44.3 ± 5.5	0.013	0.016	0.034
**Tibial nerve**	N = 142	N = 134			
Latencies (ms)	3.93 ± 0.66	4.06 ± 0.69	0.119		
Amp, mV	11.37 ± 4.15	10.75 ± 4.60	0.25		
NCV, m/s	45.3 ± 4.4	43.8 ± 4.6	0.006	0.005	0.029
F-wave					
**Median nerve**	N = 141	N = 136			
Latencies (ms)	27.5 ± 2.7	28.6 ± 3.5	0.004	0.006	0.052
**Ulnar nerve**	N = 141	N = 138			
Latencies (ms)	27.0 ± 2.7	28.3 ± 3.1	<0.001	<0.001	<0.001
**Peroneal nerve**	N = 115	N = 102			
Latencies (ms)	48.7 ± 4.7	50.6 ± 4.8	0.004	0.002	0.008
**Tibial nerve**	N = 141	N = 131			
Latencies (ms)	49.1 ± 4.4	50.8 ± 5.1	0.004	0.001	0.007
H-reflex					
**Tibial nerve**	N = 139	N = 128			
Latencies (ms)	31.6 ± 2.8	32.6 ± 3.2	0.006	0.002	0.012

Data are presented as the mean ± standard deviations (SD) for continuous values; * adjusted for age, B.W, B.H; ** adjusted for age, B.W, B.H, and HbA1c. Amp: amplitude; NCV: nerve conduction velocity.

**Table 5 jcm-11-03033-t005:** Correlation between GDF15 and nerve conduction velocity.

LogGDF15 (pg/mL)	Correlation Coefficient (r)	*p*-Value
Sensory		
**Median nerve**	N = 271	
Latencies (ms)	0.229 (0.118 *,0.062 **)	<0.001 (0.054 *,0.314 **)
Amp, µV	−0.109	0.073
NCV, m/s	−0.224 (−0.155 *,−0.083 **)	<0.001 (0.012 *,0.179 **)
**Ulnar nerve**	N = 272	
Latencies (ms)	0.152 (0.119 *,0.068 **)	0.012 (0.052 *,0.267 **)
Amp, µV	−0.147 (−0.101 *,−0.044 **)	0.016 (0.099 *,0.473 **)
NCV, m/s	−0.188 (−0.139 *,−0.095 **)	0.002 (0.024 *,0.123 **)
Motor		
**Ulnar nerve**	N = 283	
Latencies (ms)	0.178 (0.130 *,0.074 **)	0.003 (0.029 *,0.217 **)
Amp, mV	−0.075 (−0.106 *,−0.64 **)	0.207 (0.075 *,0.283 **)
NCV, m/s	−0.143 (−0.131 *,−0.045 **)	0.055 (0.080 *,0.546 **)
**Peroneal nerve**	N = 272	
Latencies (ms)	0.104	0.086
Amp, mV	−0.116	0.056
NCV, m/s	−0.138 (−0.088 *,−0.065 **)	0.023 (0.150 *,0.287 **)
**Tibial nerve**	N = 277	
Latencies (ms)	0.168 (0.199 *,0.153 **)	0.005 (0.001 *,0.011 **)
Amp, mV	−0.165 (−0.094 *,−0.060 **)	0.006 (0.121 *,0.320 **)
NCV, m/s	−0.203 (−0.198 *,−0.152 **)	0.001 (0.001 *,0.012 **)
F-wave		
**Median nerve**	N = 277	
Latencies (ms)	0.137 (0.067 *,−0.008 **)	0.022 (0.266 *,0.890 **)
**Ulnar nerve**	N = 279	
Latencies (ms)	0.192 (0.147 *,0.088 **)	0.001 (0.014 *,0.147 **)
**Peroneal nerve**	N = 217	
Latencies (ms)	0.192 (0.181 *,0.125 **)	0.005 (0.008 *,0.067 **)
**Tibial nerve**	N = 272	
Latencies (ms)	0.209 (0.263 *,0.203 **)	0.001 (<0.001 *,0.001 **)
H-reflex		
**Tibial nerve**	N = 267	
Latencies (ms)	0.205 (0.226 *,0.174 **)	0.001 (<0.001 *,0.005 **)

Data are presented as the mean ± standard deviations (SD) for continuous values; * adjusted for age, B.W, B.H; ** adjusted for age, B.W, B.H, and HbA1c.

**Table 6 jcm-11-03033-t006:** Correlation between GDF15 and clinical parameters.

LogGDF15 (pg/mL)		
	Correlation Coefficient (r)	*p*-Value
Age (years)	0.083	0.162
B.H (cm)	−0.012	0.843
B.W (kg)	0.018	0.757
BMI (kg/m^2^)	0.029	0.621
WC (cm)	0.254 (0.332 *)	<0.001 (<0.001 *)
HC (cm)	0.165 (0.339 *)	0.005 (<0.001 *)
HbA1c (%)	0.306 (0.302 *)	<0.001 (<0.001 *)
Creatinine	0.240 (0.146 *)	<0.001 (0.017 *)
eGFR (mL∙min^−1^1.73 m^−2^)	−0.275 (−0.190 *)	<0.001 (<0.002 *)
Urine ACR (mg/g)	0.318 (0.126 *)	<0.001 (0.040 *)
HOMA-IR	0.107 (0.166 *)	0.079 (0.007 *)
hsCRP (mg/L)	0.102 (0.050 *)	0.086 (0.419 *)

* Adjusted for age and BMI. B.H: body height; B.W: body weight; BMI: body mass index; WC: waist circumference; HC: hip circumference; HbA1c: glycated hemoglobin; eGFR: estimated glomerular filtration rate; ACR: albumin-to-creatinine ratio; hsCRP: high-sensitivity C-reactive protein; HOMA-IR: homeostatic model assessment for insulin resistance.

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
