# Peer review of "Circulating Growth Differentiation Factor 15 Is Associated with Diabetic Neuropathy"

_jcm, 2022, doi:10.3390/jcm11113033_

Round 1

Reviewer 1 Report

There are some improvements that should be made to the article. Since they are related to statistical analyses they don’t deny its publication, since statistical analyses can always be improved, and I believe they will.

Methods

Statistical analyses

  1. Please specify what method was used to identify the cutoff value (J, d2 …)

  2. Please specify whether you used for the linear models the assessment of: multicolinearity, heteroskedasticity, normality of residuals, assessment of leverage points, influential points, outliers, checking for linearity of each predictor to the dependent variable with special charts – i.e. component residual plots (not with simple scatter-plots that are unreliable). If not done, please assess them and use appropriate methods if they don’t apply (i.e. box cox transformations, sandwich estimators or robust regressions … removal of multicolinear variables …), and describe all that was performed.

Results

  1. Tables – Please write the text without abbreviation for B H, … or please explain those abbreviations in the legend. Please state that quantitative data is presented as mean +/- SD.

  2. Please remove the % sign in the content of the table and write it in a bracket after the variable name

  1. Table 3 – please remove GDF15 and logGDF15 since you compare two groups regarding GDF >530, thus there will be a difference …

  1. I would be very surprised if data is normally distributed even after logarithmation … for some of the variables. Why not present medians and IQRs? The means and Sds presented in the tables are after removing the logarithm, or they are without applying the logarithm? Please specify this in great detail even in the legend of the table, since it makes all the difference.

  1. Also for correlation Partial Spearman correlation could be used to adjust for other variables, if the assumption of normality does not hold for both variables for which the correlation is checked.

  1. Figures. Please put the a), and b) … on the figures.

  1. Please remove the text “Validation of nephropathy Risk Prediction Model”. There was no validation. The sample size for nephropathy is very small thus not reliable, furthermore you didn’t use another sample on which to validate the model built on the training set. Also cross-validation on such a small number of subjects won’t be very reliable. The biggest problem is that the group with the smallest number of subjects matters most in logistic regressions to know how much useful information we have at hand, not the total number of subjects.

Discussion

Study limitation

  1. Please add that the sample size for logistic regression models used for AUC or for AUC was limited due to the small number of diabetics / and especially diabetic nephropathy subjects. Please add that despite the adjustments made, residual confounding might remain due to the observational nature of the study. Furthermore, there are tons of statistical tests in the article, thus please state that spurious results might occur just by chance due to the multiple testing – alpha inflation.

Reviewer 2 Report

The authors conducted a study to investigate the predicted role of circulating GDF15 in diabetic metabolism and diabetic neuropathy. They suggest a significant association between the increased serum GDF-15 level and metabolic parameters and diabetic neuropathy, showing that plasma GDF15 may play an independent predictor of diabetic neuropathy.

The study is well conducted, and the research design, the results, and the conclusions are appropriate and well presented.
I have just one minor concern:
The Authors should speculate more about the possible mechanism by which GDF-15 is involved in diabetic neuropathy.
